# A Pleiotropic Role of Long Non-Coding RNAs in the Modulation of Wnt/β-Catenin and PI3K/Akt/mTOR Signaling Pathways in Esophageal Squamous Cell Carcinoma: Implication in Chemotherapeutic Drug Response

Uttam Sharma [1], Masang Murmu [1], Tushar Singh Barwal [1], Hardeep Singh Tuli [2], Manju Jain [3], Hridayesh Prakash [4], Tea Kaceli [5], Aklank Jain [1,*] and Anupam Bishayee [5,*]

1 Department of Zoology, Central University of Punjab, Ghudda 151 401, Punjab, India; uttamsharma1994@gmail.com (U.S.); bikasmasang@gmail.com (M.M.); tushar101singhbarwal@gmail.com (T.S.B.)
2 Department of Biotechnology, Maharishi Markandeshwar (Deemed to be University), Mullana-Ambala 133 207, Haryana, India; hardeep.biotech@gmail.com
3 Department of Biochemistry, Central University of Punjab, Ghudda 151 401, Punjab, India; manjujainmda@gmail.com
4 Amity Institute of Virology and Immunology, Amity University, Noida 201 301, Uttar Pradesh, India; hprakash@amity.edu
5 College of Osteopathic Medicine, Lake Erie College of Osteopathic Medicine, Bradenton, FL 34211, USA; tkaceli19970@med.lecom.edu
* Correspondence: aklankjain@gmail.com (A.J.); abishayee@lecom.edu or abishayee@gmail.com (A.B.)

**Abstract:** Despite the availability of modern techniques for the treatment of esophageal squamous cell carcinoma (ESCC), tumor recurrence and metastasis are significant challenges in clinical management. Thus, ESCC possesses a poor prognosis and low five-year overall survival rate. Notably, the origin and recurrence of the cancer phenotype are under the control of complex cancer-related signaling pathways. In this review, we provide comprehensive knowledge about long non-coding RNAs (lncRNAs) related to Wnt/β-catenin and phosphatidylinositol-3-kinase (PI3K)/protein kinase B (Akt)/mammalian target of rapamycin (mTOR) signaling pathway in ESCC and its implications in hindering the efficacy of chemotherapeutic drugs. We observed that a pool of lncRNAs, such as *HERES*, *TUG1*, and *UCA1*, associated with ESCC, directly or indirectly targets various molecules of the Wnt/β-catenin pathway and facilitates the manifestation of multiple cancer phenotypes, including proliferation, metastasis, relapse, and resistance to anticancer treatment. Additionally, several lncRNAs, such as *HCP5* and *PTCSC1*, modulate PI3K/Akt/mTOR pathways during the ESCC pathogenesis. Furthermore, a few lncRNAs, such as *AFAP1-AS1* and *LINC01014*, block the efficiency of chemotherapeutic drugs, including cisplatin, 5-fluorouracil, paclitaxel, and gefitinib, used for ESCC treatment. Therefore, this review may help in designing a better therapeutic strategy for ESCC patients.

**Keywords:** esophageal squamous cell carcinoma; long non-coding RNAs; Wnt/β-catenin; PI3K/Akt/mTOR; chemotherapy; signaling pathways

## 1. Introduction

Worldwide, esophageal cancer (EC) ranks eighth and sixth in terms of incidence and mortality among all cancers, respectively [1]. Despite the advancement in diagnostic and therapeutic applications, the overall survival of esophageal squamous cell carcinoma (ESCC) patients is still meager. For example, the five-year survival rate of ESCC patients in several less developed countries is very low (~10%), whereas in developed countries, such as the United States, the five-year survival rate is ~18% [2]. Although chemotherapy and

radiotherapy can increase the disease-free and overall survival among ESCC patients, tumor cells adopt the tendency to resist the effect of chemotherapeutic drugs or radiation doses, suggesting the development of therapy resistance mechanisms in tumor cells [3]. According to a previous report, chemotherapeutic drug resistance leads to more than 90% of deaths in patients with ESCC [4]. This may be due to the cross-networking of the vital biological, molecular, and cellular signaling pathways, such as Wnt/β-catenin, phosphatidylinositol-3-kinase (PI3K)/protein kinase B (Akt)/mammalian target of rapamycin (mTOR) signaling pathways, with the various chemotherapeutic drugs [4], thus limiting the efficacy of therapies, resulting in poor prognosis, tumor metastasis, and recurrence [5]. Therefore, understanding the underpinnings that regulate the vital signaling pathways and resist the efficacy of the chemotherapeutic drugs, is urgently required.

In recent years, improved knowledge of oncology research has led to the identification of non-coding RNAs (ncRNAs) that regulate tumor cell proliferation, differentiation, angiogenesis, metastasis, and invasion. ncRNAs include structural and regulatory RNAs, representing ~90% of the human genome. Structural long non-coding RNAs (lncRNAs) include ribosomal RNAs and transfer RNAs, whereas regulatory RNAs include small conditional RNAs, small nucleolar RNAs, microRNAs (miRNAs), and lncRNAs [6,7]. Among them, lncRNAs having size ≥200 nucleotides are involved in various biological, molecular, and cellular processes, such as transcription, splicing, translation, protein localization, epigenetics, cell structure integrity, cell cycle, cell fate determination, cell differentiation, cell migration, and cell proliferation [8]. Furthermore, lncRNAs have been implicated in modulating various cancer-related signaling pathways, such as Wnt/β-catenin [9–17], PI3K/Akt/mTOR [18–20], Janus kinase/signal transducers and activators of transcription (JAK/STAT3) [21,22], mitogen-activated protein kinase 1 (MAPK) [23,24], nuclear factor-κB (NF-κB) [25,26], and NOTCH [27–29], and display the cancer phenotypes [9–29]. Additionally, an explosion of research revealed that lncRNAs act as a mediator in regulating chemoresistance by altering the efflux of a drug, DNA damage repair, inhibition of apoptosis, and mutation of the drug targets [3,30]. Furthermore, lncRNAs play an important role in conferring radioresistance in ESCC, as documented previously by our group [3].

Current research advancements in clinical oncology revealed that lncRNAs play a vital role in cancer therapeutics, diagnosis, and prognosis. Interestingly, ESCC attracted the interest of oncologists due to its delayed diagnosis and vast number of annual deaths. Based on this idea, we searched in PubMed with a combination of keywords: lncRNA; long non-coding RNA; and esophageal squamous cell carcinoma. We obtained a pool of lncRNAs evolved from 2012–2021 in ESCC, evidenced by the increasing number of research papers appearing in PubMed (Figure 1 and Supplementary Table S1). As a result, we found that ESCC-associated lncRNAs, such as *LINC01014* [18], *HCP5* [19], and *PTCSC1* [20], modulate the PI3K/Akt/mTOR pathway. In addition, lncRNAs, namely *LOC146880* [23], *BANCR* [31], and *LINC00324* [24], are involved in MAPK signaling pathways. Similarly, studies reported that dysregulation of other lncRNAs (shown in Supplementary Table S1) dysregulate hedgehog [32], p53 [33], NF-κB [25,26], NOTCH [27,28], TGFβ1 [34], STAT [21,22], and Wnt/β-catenin signaling [9–17] in ESCC. Among all the signaling pathways, we identified most of the lncRNAs modulating the Wnt/β-catenin and PI3K/Akt/mTOR signaling pathways in ESCC, suggesting the implication of these two pathways during ESCC pathogenesis.

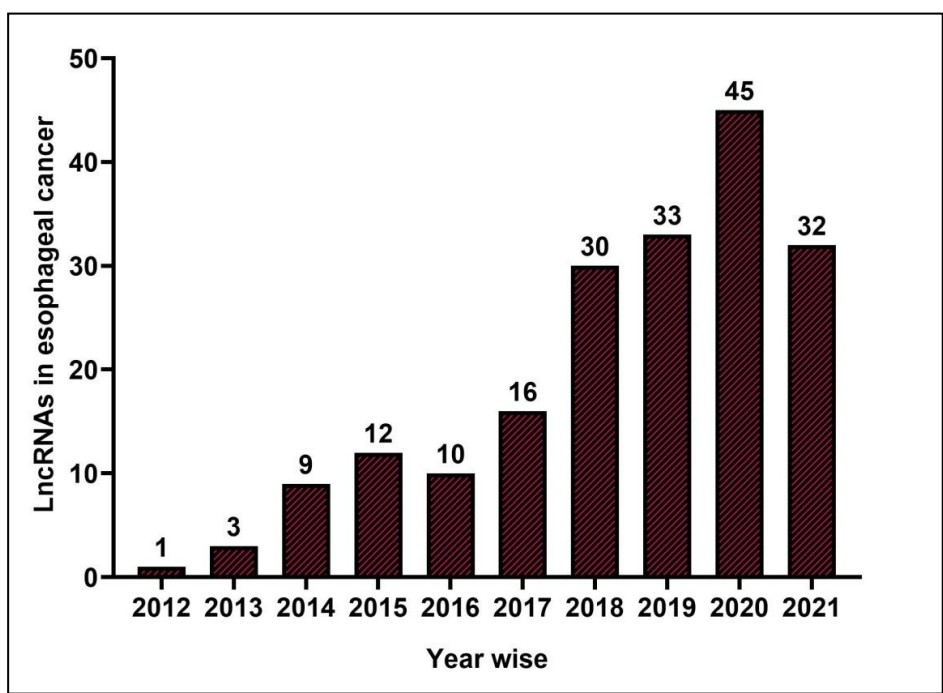

**Figure 1.** The bar graph represents the trends of lncRNAs evolution with year.

Previous studies highlighted the role of lncRNAs with Hippo, transforming growth factor beta (TGFβ)/SMAD, and JAK/STAT signaling pathways [7,35,36] but did not summarize the detailed association with Wnt/β-catenin and PI3K/Akt/mTOR in ESCC. Therefore, the objective of this work was to analyze the regulatory role of Wnt/β-catenin and PI3K/Akt/mTOR pathways in association with lncRNAs in ESCC and their role in chemotherapeutics drug response. Additionally, we have presented the crosstalk between the Wnt/β-catenin and PI3K/Akt/mTOR signaling pathway in ESCC. Thus, our review provides comprehensive knowledge about the underpinnings that need to be targeted to better the treatment of ESCC patients.

## 2. Wnt/β-Catenin Signaling Pathway-Related lncRNAs in ESCC

The Wnt signaling pathway is a well-known, evolutionarily conserved pathway that regulates cell proliferation, migration, and invasion and thus controls tumor progression [37]. Genetic and epigenetic alterations, such as DNA hypermethylation in the promoter region of axis inhibition protein 2 (Axin2), adenomatous polyposis coli (*APC*), Wnt inhibitory factor 1 (*WIF-1*), and secreted frizzled-related protein (*SFRPs*), lead to the aberrant activation of Wnt/β-catenin signaling pathway in several types of tumors, including ESCC [14]. Based on the signal transduction mechanism, Wnt signaling is classified into canonical and non-canonical pathways. Canonical Wnt signaling translates the transcriptional activator β-catenin into the nucleus, and constitutive activation leads to cancer pathogenesis. In contrast, non-canonical Wnt pathways are independent of β-catenin transcriptional activity and hence regulated via Wnt polarity, Wnt-$Ca^{2+}$, and Wnt-atypical protein kinase signaling (Figure 2).

In cancer cells, several lncRNAs, such as highly expressed lncRNA in esophageal squamous cell carcinoma (*HERES*), small nucleolar RNA host gene 16 (*SNHG16*), urothelial cancer associated 1 (*UCA1*), maternally expressed 3 (*MEG3*), *LINC00675*, HOX antisense intergenic RNA (*HOTAIR*), taurine upregulated gene 1 (*TUG1*), and growth-arrest-associated long non-coding R/NA (*GASL1*), target and affect the expression of β-catenin, a pivotal molecule of the Wnt signaling pathway, which regulates the expression of Wnt target genes and the function of cancer cells in Wnt/β-catenin signaling pathway. These lncRNAs have been observed to play a pivotal role in Wnt/β-catenin signaling modulation in various

cancers, including ESCC (Figure 3). Interestingly, Wnt/β-catenin pathway-related lncRNAs can directly or indirectly stimulate various subunits of the Wnt/β-catenin pathway, thereby activating or inhibiting the pathway's activity. Therefore, understanding Wnt signaling in the context of lncRNAs may be a valuable strategy for managing ESCC.

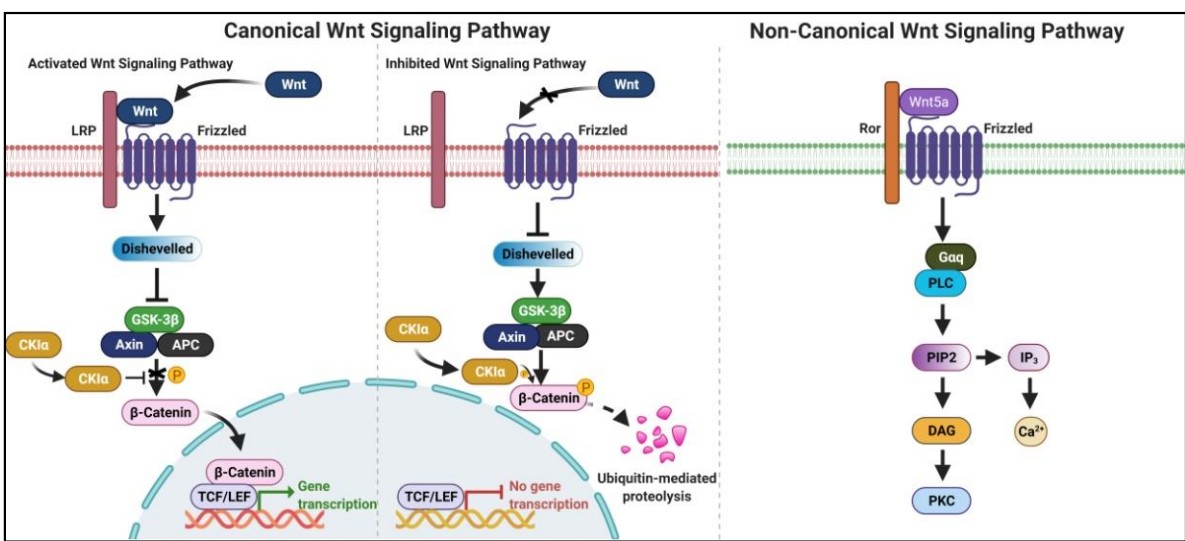

**Figure 2.** Role of Wnt/β-catenin signaling pathway in cancer. In the activated canonical pathway. Wnt allows the connection between Frizzled receptor and lipoprotein receptor-related protein (LRP), which further activates the Dishevelled followed by inhibition of glycogen synthase kinase 3 (GSK-3β), axis inhibitor (AXIN), adenomatous polyposis coli (APC) and cyclin-dependent kinase inhibitor (CKIα) complex. This complex inhibits the phosphorylation of β-catenin, subsequently enters into the nucleus, and transcribes the cancer-related genes with the help of the TCF/LEF complex. The mechanism is vice-versa in the inhibited canonical Wnt pathway. The non-canonical Wnt pathway utilizes Wnt5a for the activation of the pathway and allows the gene transcription through calcium ions. This illustration was created using resources available at www.biorender.com (accessed on 25 September 2021).

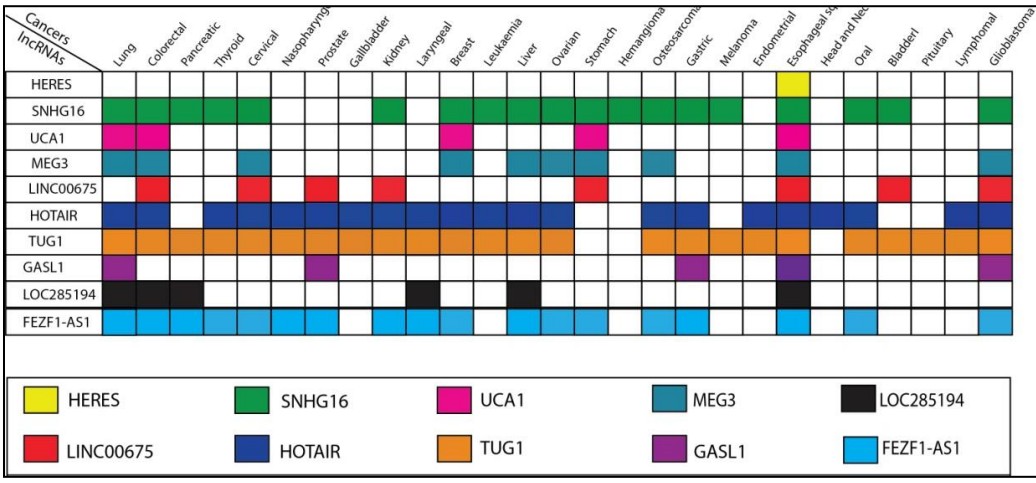

**Figure 3.** Dysregulation profile of lncRNAs in various cancers including ESCC irrespective of signaling pathways. Each colored box represents the association of lncRNAs with various cancers.

For instance, *HERES* showed upregulation in 66 ESCC tissues compared to adjacent non-cancerous tissue samples [9]. lncRNA *HERES* augments five Wnt signaling regulated genes viz, calcium voltage-gated channel auxiliary subunit alpha2 delta 3 (*CACNA2D3),*

secreted frizzled related protein 2 (*SFRP2*), calcium voltage-gated channel subunit alpha1 E *(CACNA1E)*, CXXC finger protein 4 *(CXXC4)*, and secreted frizzled related protein 2 (*SFRP4)* [9]. *CACNA2D3* encodes a Wnt/Ca$^{2+}$ complex subunit by decreasing intracellular calcium levels and the expression of Nemo-like kinase (NLK). Furthermore, downregulation of HERES increases NLK protein expression and reduces the β-catenin levels in KYSE-70 and HCE-7 ESCC cell lines (Figure 4). Additionally, *SFRP2* encodes a member of the SFRP family that regulates the Wnt signaling pathway [9]. Furthermore, *SFRP2* and *CXXC4* act as negative regulators of the canonical Wnt signaling pathways. Thus, taken together, HERES promotes the ESCC pathogenesis by targeting both canonical and non-canonical pathways.

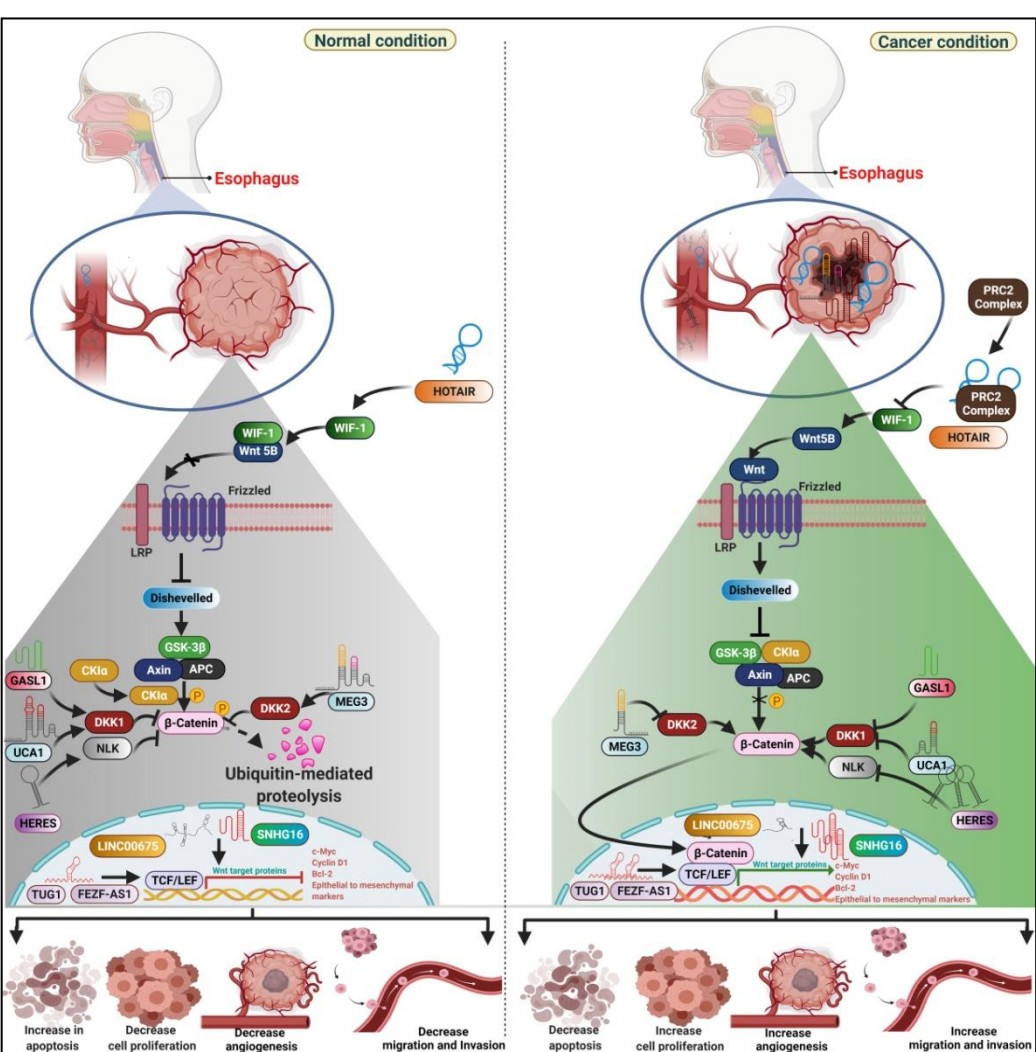

**Figure 4.** Mechanism of Wnt/β-catenin related lncRNAs in normal and tumorigenic conditions. LncRNA *HOTAIR* in association with PRC2 complex inhibits the WIF-1 expression, and thus Wnt5B becomes free and allows the binding of the Frizzled receptor and lipoprotein receptor-related protein (LRP). As a result, β-catenin does get not phosphorylated and eventually enters the nucleus and transcribes the c-Myc, cyclin D1, Bcl-2, EMT genes. Furthermore, low expression of *MEG3* inhibits the Dickkopf-2 (DKK2), whereas, *GASL1* and *UCA1* inhibit the Dickkopf-1 (DKK1). In addition, *HERES* inhibits the expression of Nemo Like Kinase (NLK). As a result, an enormous amount of β-catenin is generated and transcribe the c-Myc, cyclin D1, Bcl-2, EMT genes. At the same time, another set of lncRNAs Taurine Up-Regulated 1 (*TUG1*), *LINC00675*, Small Nucleolar RNA Host Gene 16 (*SNHG16*) localized in the nucleus and executes their action on cancer-related proteins. This illustration was created using resources available at www.biorender.com (accessed on 25 September 2021).

Similarly, lncRNA *SNHG16* levels were significantly upregulated in ESCC tissues compared to normal tissue samples [10]. lncRNA *SNHG16* promotes cell proliferation and invasion by modulating the targets of the Wnt/β-catenin pathway. In line with this, the TOPFLASH and FOPFLASH (TOP/FOP) luciferase reporter system showed that knockdown of *SNHG16* expression inhibited the activation of the Wnt/β-catenin signaling in ESCC cell line EC-1 and Eca-109 [10]. Subsequently, the expression of regulatory molecules of Wnt signaling, such as c-Myc, β-catenin, and cyclin D1, was markedly reduced in the *SNHG16* knockdown cell line, suggesting *SNHG16* could be one of the markers to detect the alternation of the Wnt/β-catenin signaling pathway [10] (Figure 4). In addition to *SNHG16*, *FEZF1-AS1* was significantly upregulated in 45 pairs of ESCC tissues and cells compared to adjacent non-neoplastic tissues and Het-1A cells, respectively, using real-time quantitative reverse transcription-polymerase chain reaction (qRT-PCR) [38]. lncRNA *FEZF1-AS1* promoted the migration and invasion of ESCC cells but did not affect the cell proliferation and cell cycle of ESCC cells. This phenotype is ascertained by the overexpression of Wnt regulated protein β-catenin in ESCC tissues [38].

In addition to the above, lncRNA *UCA1* acts as a tumor suppressor with low expression in 106 ESCC tumor samples compared to adjacent normal tissues [11]. Through bioinformatics analysis, the authors suggested that *UCA1* regulates Wnt signaling downstream molecules, such as catenin β1 (*CTNNB1*), dishevelled associated activator of morphogenesis 2 (*DAAM2*), dickkopf wnt signaling pathway inhibitor 1 (DKK1), and Wnt family member 2B (*WNT2B*) (Figure 4) [11]. Further overexpression of *UCA1* leads to a reduction in the expression of the c-Myc target gene of Wnt signaling and thus regulates the cell cycle, suggesting that overexpression of *UCA1* inhibits the activity of the Wnt/β-catenin signaling pathway in EC109 cells (Figure 4) [11]. Interestingly, the tumor suppressor *MEG3* promotes tumor progression through targeting miR-4261, modulating the expression of DKK2, β-catenin, Bcl-2, and c-Myc, thus activating the Wnt/β-catenin signaling pathway [12] (Figure 4). As a result, *MEG3* enhances the proliferation, migration, and invasion of ESCC cells [12]. Not only *MEG3,* but lncRNA *LINC00675* was also downregulated in ESCC tissue samples compared to matched normal tissues [13]. Ectopic expression of *LINC00675* reduced cell proliferation, migration, and invasion by decreasing cell cycle proteins, such as cyclin D1 and c-Myc, and epithelial-mesenchymal transition (EMT) regulated proteins, such as N-cadherin and vimentin, in EC9706 and EC-1 cells [13] (Figure 4). The above mechanism was occurred by targeting the β-catenin, a vital molecule of the Wnt/β-catenin signaling pathway, which suggests that enhancing the expression of *LINC00675* inhibits the activities of the Wnt/β-catenin signaling pathway in ESCC cells [13]. lncRNA *HOTAIR* is frequently detected as oncogenic in ESCC patients' tissues and is associated with the Wnt/β-catenin signaling pathway subunits. The upregulated profile of *HOTAIR* in ESCC tissues and cell lines targets an essential regulatory molecule of the Wnt pathway, WNT5B, and WIF-1 [14]. WIF-1 acts as a key inhibitor of the Wnt/β-catenin signaling pathway. It facilitates the degradation of β-catenin via APC/Axin1 destruction complex and by preventing the interaction of extracellular Wnt ligands with their receptors [14] (Figure 4). Mechanistically, epigenetic silencing of WIF-1 causes altered activation of the Wnt/β-catenin pathway in ESCC. Notably, WIF-1 downregulation is a prominent marker of tumor progression. qRT-PCR data revealed that the *HOTAIR* overexpression decreases the protein levels of WIF-1 and thus alters the Wnt/β-catenin signaling [14] (Figure 4). Moreover, *HOTAIR* exerts its function via a PRC2-dependent mechanism. The depletion of PRC2 enhances the levels of WIF-1 mRNA [14]. Additionally, an immunoblot assay revealed *HOTAIR* overexpressed ESCC cells possess a higher concentration of β-catenin expression in the nucleus, which indicates the activation of the canonical Wnt/β-catenin pathway [14]. Taken together, PRC2-associated *HOTAIR* inhibits the expression of WIF-1 by increasing trimethylation at H3K27 in the WIF-1 promoter region and then activates the Wnt/β-catenin signaling pathway and manifests the cell proliferation, migration, and invasion of ESCC cells (Figure 4).

At the same time, lncRNA *TUG1* contributes to tumor progression by overexpressing their levels in 40 ESCC patients' tissues and cell lines EC9706 and OE19tumor-adjacent corresponding tissues and HEEC cells, respectively. Upregulated *TUG1* exerts its potential effects on ESCC manifestation through enhancing Wnt/β-catenin pathway-associated protein markers, such as Wnt1, c-Myc, cyclinD1, and β-catenin [15] (Figure 4). Furthermore, the administration of an activator (SKL2001) or inhibitor (XAV939) of Wnt/β-catenin signaling pathway to the *TUG1*-knockout ESCC cell lines EC9706 and OE19 revealed that SKL2001 promoted the expression of N-cadherin, Vimentin, and Snail and abolished the expression of E-cadherin, and thus enhanced the migration and invasion of the ESCC cells [15]. In addition to EMT, SKL2001 accelerated cell viability and cell apoptosis. Moreover, reverse effects were observed in XAV939 administered *TUG1*-knockout ESCC cell line, which suggests that the upregulation profile of *TUG1* activates the Wnt/β-catenin signaling pathway [15], thus enhancing the proliferation, migration, and invasion and diminishing the apoptosis of ESCC cells. Another set of tumor suppressor lncRNA *GASL1* manifests the ESCC pathogenesis by regulating the subunits of Wnt/β-catenin signaling pathway subunits. Downregulated *GASL1* levels increase the protein expression of Wnt3a, β-catenin, and c-Myc and decrease the protein expression of DKK1 [16] (Figure 4), which suggests the activation of the canonical Wnt/β-catenin signaling pathway and ultimately the enhancement of ESCC cell proliferation, migration, and invasion.

Overall, we can say that the above-mentioned Wnt/β-catenin signaling pathway-related lncRNAs contribute to developing a therapeutic target in treating ESCC. Therefore, inactivation of the Wnt/β-catenin signaling pathway through altering the levels of lncRNAs could be effective in treating ESCC patients. Moreover, intrinsic and acquired resistance may limit the therapeutic efficacy of Wnt/β-catenin signaling pathway inhibitors

## 3. PI3K/Akt/mTOR Pathway-Related lncRNAs in ESCC

PI3K, Akt, and mTOR are the three significant nodes in the PI3K/Akt/mTOR pathway [39,40]. Tyrosine kinases and other receptor molecules, such as growth factors, hormones, and mitogen stimuli, activate the PI3K, Akt, and mTOR [39]. The PI3K/Akt/mTOR signaling pathway is one of the most critical intracellular pathways regulating cell growth, proliferation, metabolism, motility, survival, and apoptosis [41]. Therefore, aberrant activation of the PI3K/Akt pathway contributes to the development of tumor PI3K. As a result, PI3K promotes the survival and proliferation of tumor cells in many human cancers [42–45]), including ESCC (Figure 5). Recently, it has been reported that lncRNAs, such as HLA Complex P5 (*HCP5*), Papillary Thyroid Carcinoma Susceptibility Candidate 1 (*PTCSC1*), and *LINC01014*, and the PI3K/Akt/mTOR pathway are in tight conjunction during ESCC pathogenesis. This emphasizes the need to target this pathway with associated lncRNAs in treating ESCC patients.

lncRNA *HCP5* was upregulated in ESCC tissues compared to control tissues [19]. Additionally, lncRNA HCP5 promotes proliferation, migration, invasion ability, and stemness characteristics of ESCC cells. Furthermore, it suppresses ESCC cell apoptosis by sponging miR-139-5p, thus upregulating phosphodiesterase 4A (*PDE4A*), a downstream target gene of the PI3K/Akt/mTOR pathway [19] (Figure 6). Additionally, lncRNA *PTCSC1* expression was elevated in ESCC tissues and cells compared to adjacent non-cancerous tissues [20]. As a result, *PTCSC1* promotes cell proliferation, migration, and invasion by activating the Akt p85 subunit of the PI3K/Akt/mTOR pathway [20]. In line with this, phosphorylated Akt levels also increased in *PTCSC1* overexpressed KYSE30 cells [20], suggesting that *PTCSC1* activated Akt signaling in ESCC cells (Figure 6). Last but not least, *LINC01014* is associated with the PI3K/Akt/mTOR pathway in relation to gefitinib drug resistance in ESCC [18]. However, a detailed study in the context of this topic has not been elucidated in detail yet.

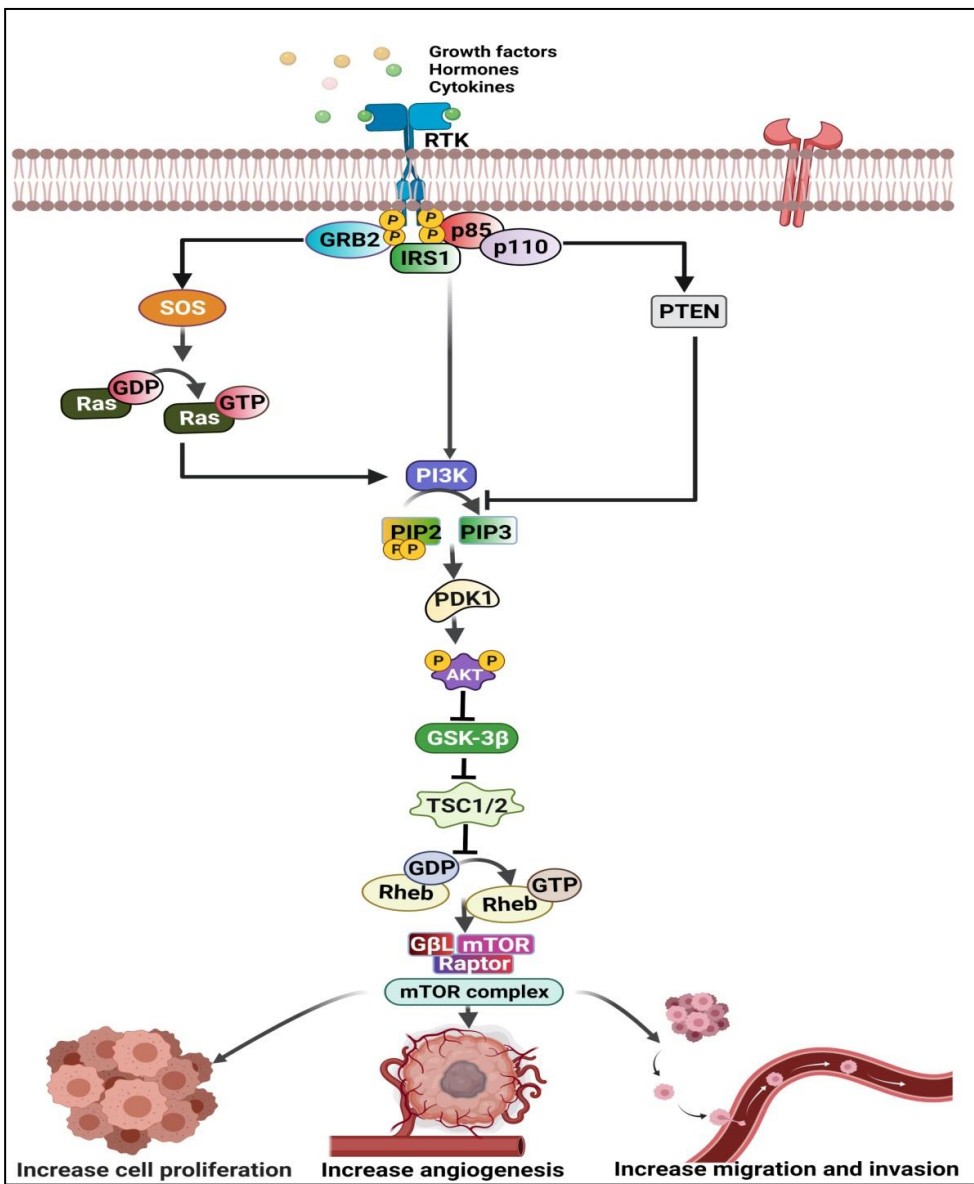

**Figure 5.** Mechanism of PI3K/Akt/mTOR pathway in cancer. The tyrosine kinase (RTK) receptor becomes activated by growth factors, hormones, and cytokines, which activates the p85 subunit, Ras and phosphatase and tensin homolog (PTEN). As a result, GSK-3β becomes inhibited, which further activates the mTORC1 complex and thus manifests the hallmarks of cancer. This illustration was created using resources available at www.biorender.com (accessed on 25 September 2021).

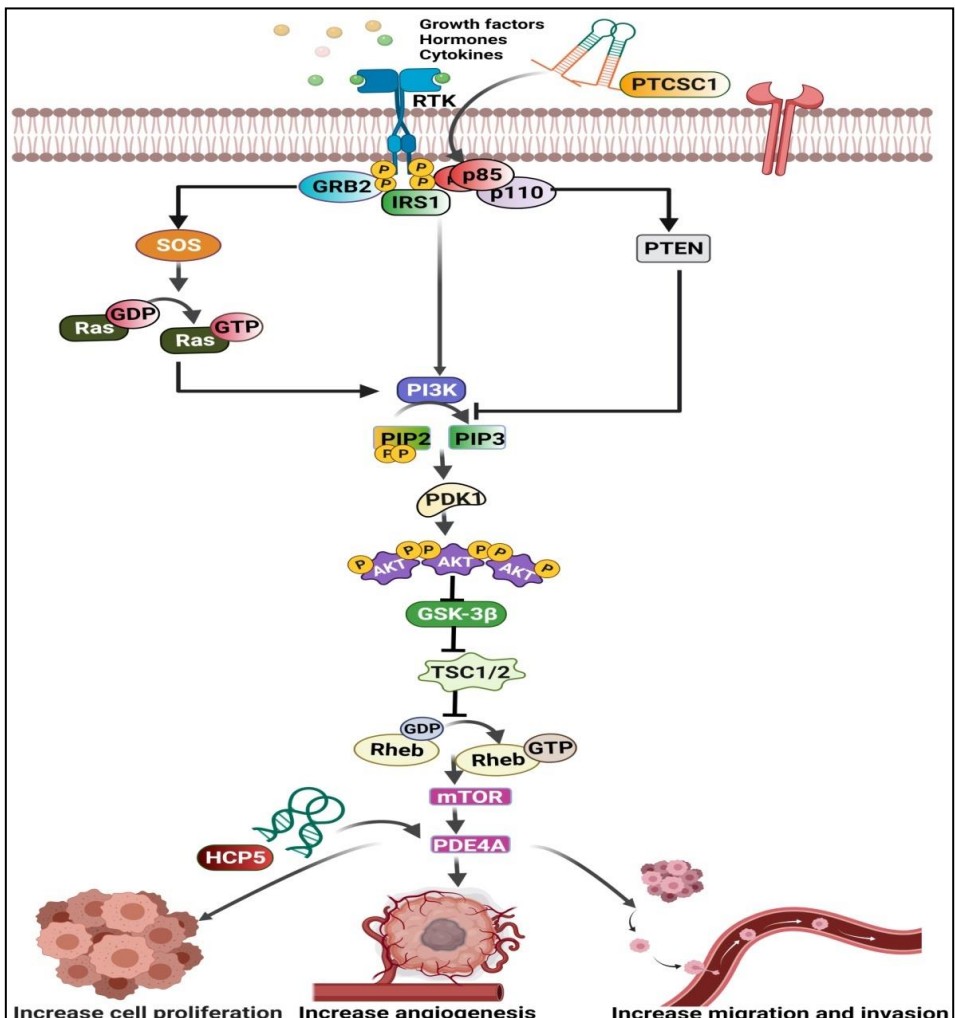

**Figure 6.** Interaction of lncRNAs with PI3K/Akt/mTOR pathway in ESCC. Long non-coding RNA Papillary Thyroid Carcinoma Susceptibility Candidate 1 (*PTCSC1*) upregulates the p85 subunit and thus phosphorylate the Akt and increases the mechanistic target of rapamycin kinase (mTOR) levels followed by an increase in phosphodiesterase 4A (PDE4A) expression. At the same time, lncRNA *HCP5* acts on the PDE4A and further increases its level, enhancing the cell proliferation, angiogenesis, migration, and invasion of ESCC cells. This illustration was created using resources available at www.biorender.com (accessed on 25 September 2021).

## 4. Crosstalk between Wnt/β-Catenin and PI3K/Akt/mTOR Pathway in ESCC

In the previous section, we discussed the regulatory role of Wnt/β-catenin and PI3K/Akt/mTOR related lncRNAs during ESCC pathogenesis. Studies suggested that Wnt/β-catenin and PI3K/Akt/mTOR pathways regulate themselves via a feedback mechanism, thus representing the resistance potential to chemotherapeutic drugs in clinical settings [46]. Therefore, understanding the crosstalk between the two mentioned pathways in ESCC is of immense importance. These pathways are finely connected at multiple levels during the homeostasis and pathological condition. For instance, glycogen synthase kinase 3β (GSK3β) is identified as a common key element in both the signaling pathways and thus regulates different cellular processes (Figure 7) [47]. During the activation of both signaling pathways, GSK3β activity becomes inhibited via various upstream events. Furthermore, a fraction of AXIN-bound GSK3β targets β-catenin degradation through the phosphorylation of β-catenin. At the same time, activated PI3K phosphorylates Akt at Ser9 residue, which further inhibits the GSK3β activity (Figure 7) [46].

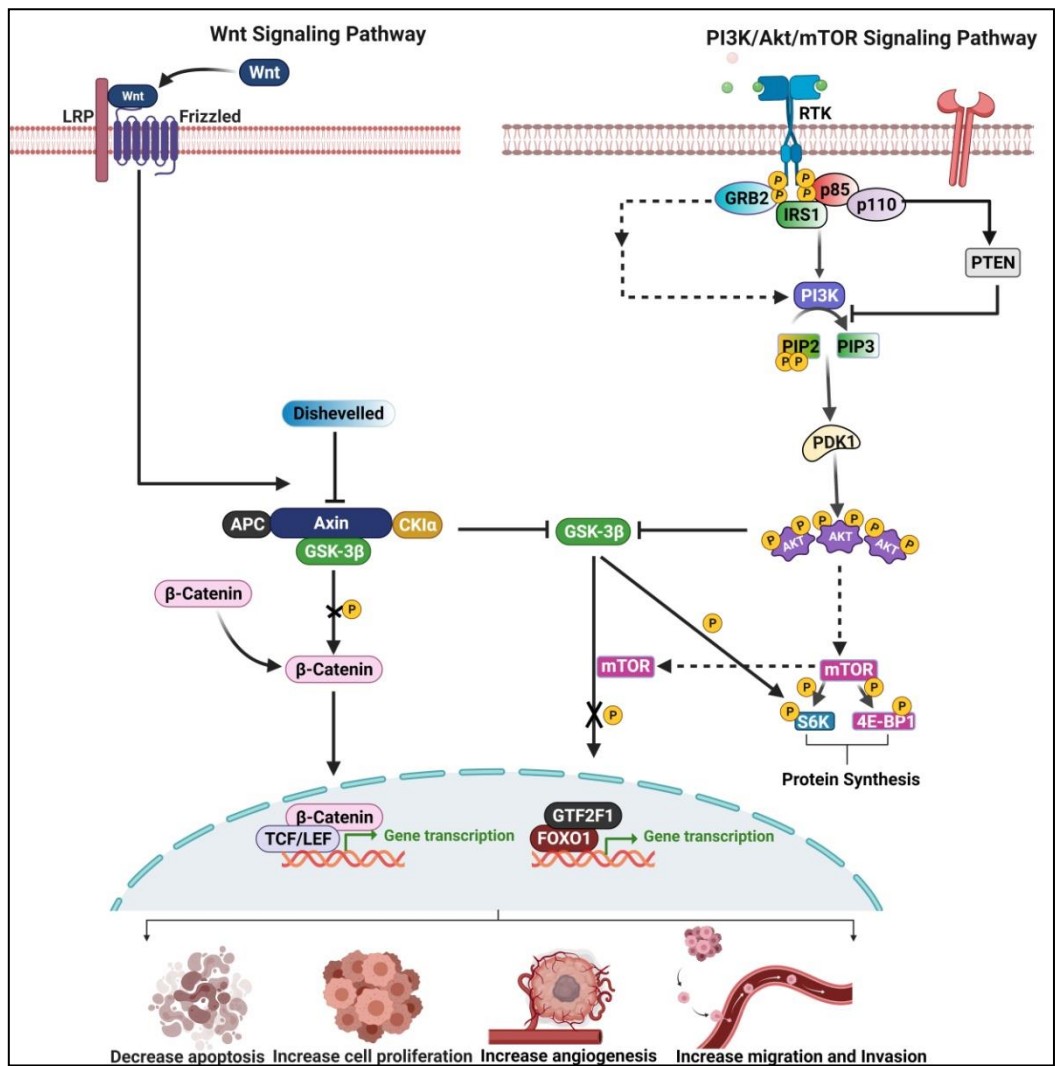

**Figure 7.** Crosstalks between Wnt/ β-catenin and PI3K/Akt/mTOR pathway. The activation of both signaling pathways results in the inhibition of glycogen synthase kinase-3β (GSK-3β) activity via various upstream events. In the Wnt/β-catenin pathway, a fraction of AXIN-bound GSK3β has a vital role in controlling β-catenin degradation through the regulation of β-catenin phosphorylation. At the same time, activated PI3K phosphorylate Akt at Ser9 residue, which further inhibits the GSK3β activity. As a result, transcription of oncogenes was initiated. This illustration was created using resources available at www.biorender.com (accessed on 25 September 2021).

At the same time, Akt hyperactivation and active canonical Wnt signaling pathways inhibit the GSK3β activity, resulting in the accumulation of β-catenin, thus representing the ESCC cell proliferation, migration, and invasion [46]. Recent studies showed that Wnt pathway activation leads to phosphorylation of S6K and 4E binding protein 1 (4E-BP1) and may affect the protein synthesis, thus turning on the mTOR complex 1 cascade (Figure 7) [48]. Notably, the activation of the Wnt-driven mTORC1 signaling could be independent of β-catenin and mediated by APC-AXIN-GSK3β axis and tuberous sclerosis 2 protein (TSC2) [48]. Moreover, it has recently been proposed that nuclear translocation of GSK3β is facilitated by rapamycin (mTOR) via increased phosphorylation of the forkhead box O1 (FOXO1) and general transcription factor IIF subunit 1 (GTF2F1), decreasing cell proliferation (Figure 7) [49,50].

## 5. LncRNAs Regulate the Efficacy of Chemotherapeutic Drugs in ESCC

Worldwide, esophageal tumors develop resistance to chemotherapy during treatment, leading to multiple fatal complications implicated in treatment failure and tumor relapse [51]. Generally, chemotherapy represents the frontline treatment for both early and advanced staged tumors. However, chemotherapeutic drug resistance limits the efficacy of conventional chemotherapeutics and the United States Food and Drug Administration-approved biological agents, such as keytruda (pembrolizumab) nivolumab, opdivo (nivolumab), and pembrolizumab. Notably, multidrug resistance in ESCC patients can be due to the higher expression of transporters that eject drugs from cells [52]. It can be intrinsic (tumor insensitive to therapeutic agents before treatment) or extrinsic (tumor becomes resistant during the treatment) [52]. However, the mechanisms associated with drug resistance in ESCC patients, including resistance to apoptosis induced by drugs, decreased intracellular accumulation of therapeutics, increased repair of damaged DNA, and induction of mechanisms capable of drug detoxification, are still in their infancy [53]. Furthermore, due to the limitations associated with the drug potency in ESCC patients, scientists aim to develop/identify new biomarkers to assess and predict patients' responses against the chemotherapeutic drugs. Recent studies showed that lncRNAs play important roles in regulating the chemo- and radio-resistance of ESCC by controlling several signaling pathways and modulating the mechanisms associated with the cell cycle, proliferation, apoptosis, and DNA damage repair [3].

With a view to the challenges faced by the ESCC patients during chemotherapy, we tried our best to sort the relevant drug molecules, which need to be studied in the context of lncRNA and ESCC for disease management. Firstly, we searched in PubMed all recent studies (published from 2012 onwards) investigating lncRNAs in chemoresistance during ESCC therapy. The primary screening result was manually curated to avoid and remove articles with generic statements and not direct links between lncRNAs and drugs in the context of ESCC. Thus, we have chosen the research paper showing the direct association of lncRNAs with chemoresistance in ESCC treatment for this section. Furthermore, we searched the reported drugs in the NoncoRNA database for collecting all information about lncRNA-target gene drugs used for ESCC therapy in association with lncRNAs.

Mounting evidence suggests that a pool of lncRNAs (*LOC285194*/ (tumor suppressor candidate 7 (*TUSC7*), taurine upregulated 1 (*TUG1*), AFAP1 antisense RNA 1 (*AFAP1-AS1*), prostate androgen-regulated transcript 1 (*PART1*), colon cancer-associated transcript 1 (*CCAT1*), long intergenic non-protein coding RNA 1419 (*LINC01419*), long intergenic non-protein coding RNA 337 (*LINC00337*), long intergenic non-protein coding RNA 1014 (*Linc01014*), MACC1 antisense RNA 1 (*MACC1-AS1*), FOXD2 adjacent opposite strand RNA 1 (*FOXD2-AS1*)) are involved in ESCC chemotherapy resistance.

It was observed that lncRNA *LOC285194* or *TUSC7* downregulated in ESCC tissues and cell lines compared to adjacent normal tissues [54]. Moreover, downregulated *LOC285194* hinders the potential of cisplatin (20 mg/m$^2$/day, for five days in combination with radiotherapy (40 Gy in 20 fractions of 2 Gy each, with five fractions per week for four weeks) (Table 1). Kaplan–Meier survival analysis revealed that low expressed *LOC285194* ESCC patients group showed decreased disease-free survival (DFS) and overall survival compared to high expressed *LOC285194* group. The complete pathological response (pathCR) rate was 57% in the *LOC285194*-high group, while only 15% in the *LOC285194*-low group suggested that patients with low expression *LOC285194* showed resistance to chemoradiotherapy treatment (Table 1) [54]. At the same time, lncRNA *TUSC7* was downregulated in the chemotherapy resistance patients' group compared to the chemotherapy responsive patients' group, and thus the survival rate of the ESCC patients became very poor. The low expression of *TUSC7* resists the potency of cisplatin (1, 2, 4, 8, and 16 µM for 48 h) or 5-FU (1, 4, 16, 32, and 64 µM for 48 h) in ESCC cell lines EC9706 and KYSE30 [55] (Table 1). In addition to that, *TUG1* is significantly upregulated in ESCC tissues compared to paired adjacent normal tissues [56]. Furthermore, *TUG1* expression is higher in TE-1 derived cisplatin (DDP)-resistant (TE-1/DDP) cells (1 µg/mL for 48 h)

compared to TE-1 cells [57], suggesting that high *TUG1* expression was significantly implicated with chemotherapy resistance and inversely correlated with overall survival of ESCC patients [56] (Table 1). Moreover, lncRNA *AFAP1-AS1* showed upregulation (~14-fold) in paired cisplatin-resistant (KYSE30-R) and parental ESCC cell lines (KYSE30) [58]. Moreover, an upregulation profile of *AFAP1-AS1* was observed in 162 pretreatment biopsy specimens of ESCC who underwent definitive chemoradiotherapy (dCRT). Notably, up-regulated *AFAP1-AS1* undergoes cross-resistance of cisplatin (0.3125, 0.625, 1.25, 2.5 5, 10, 25, and 50 μM for 24 h on days 1–4) along with two combinations of anticancer drugs viz, 5-fluorouracil (5-FU) (2, 4, 8, 16, 32, 64,128, and 256 μM for 24 h on days 1–4) and paclitaxel (0.03125, 0.0652. 0.125, 0.25, 0.5, 1, 2, 4, and 8 μM for 24 h on days 1–4), when administered to ESCC patients [58] (Table 1). Chemotherapeutics treated patients represent low overall survival and progression-free survival. Moreover, the pathological complete response rate was 19.8%, the partial response rate was 40.7%, no response rate was 37.7%, and progressive disease response was 1.8%, suggesting the strong hindrance property of lncRNA *AFAP1-AS1* in conferring the chemotherapy in ESCC management [58] (Table 1). In line with this, high expression levels of *AFAP1-AS1* serve as a potential biomarker to predict tumor response and survival.

**Table 1.** Characteristics of lncRNAs in clinical studies.

| LncRNA | Expression Pattern (Up/Down Regulation) | Drug | Conc. of Drugs Used | Time Points of Treatment | Patient Tissue/Cell Line/In Vivo Model | Clinical Endpoint | Pathological Response | Cohort Size | References |
|---|---|---|---|---|---|---|---|---|---|
| LOC285194 | Down | Cisplatin | NA | NA | Tissue and cell lines | DFS and OS | CR = 15% | Female = 48 / Male = 94 | [54] |
| TUG1 | Up | Cisplatin | 1 μg/mL | 48 h | Tissue and cells | OS | NA | Male = 171 / Female = 47 | [56] [57] |
| AFAP1-AS1 | Up | 5-Fluorouracial Cisplatin Paclitaxel | 2, 4, 8, 16, 32, 64, 128, 256 μM 0.3125, 0.625, 1.25, 2.5 5, 10, 25, 50 μM 0.03125, 0.0652, 0.125, 0.25, 0.5, 1, 2, 4, 8 μM | 24 h on days 1–4 | Tissue and cells | OS and PFS | CR = 19.8% PR = 40.7% NC = 37.7% PD = 1.8% | Male = 123 Female = 39 | [58] |
| PART1 | Up | Gefitinib | 0.01–10 μM | 48 h | Serum | NA | NA | 79 | [63] |
| TUSC7 | Down | Cisplatin 5-Fluorouracial | 1, 2, 4, 8, 16 μM 1, 4, 16, 32, 64 μM | 48 h | Tissue and cell lines | OS | NA | Male = 43 Female= 19 | [55] |
| CCAT1 | Up | Cisplatin | 0.1, 0.2, 0.5, 1, 2, 5 μM | 48 h | Cell lines | NA | NA | NA | [59] |
| LINC01419 | Up | 5-fluorouracil | 10 μg/mL | 48 h | Tissue and cell lines | NA | NA | 76 | [60] |
| LINC00337 | Up | Cisplatin | 0.5, 1, 2, 3 μg/mL | 48 h | Tissue and cell lines | NA | NA | Male = 48 Female = 26 | [51] |
| Linc01014 | Up | Gefitinib | 10 μM | 48 h | Cell lines | NA | NA | NA | [18] |
| MACC1-AS1 | Up | Cisplatin | 20, 40, 60, 80, 100 μM | NA | Tissue and cell lines | NA | NA | Male = 62 Female = 8 | [61] |
| FOXD2-AS1 | Up | Cisplatin | 20, 40, 60, 80, 100 μM 6.25, 12.5, 25, 50, 100 μg/mL | NA | Tissue and cell lines | NA | NA | Male = 62 Female = 8 | [61] [62] |

Furthermore, higher expression of *CCAT1* slightly decreases the viability of cisplatin-resistant ESCC cells (0.1, 0.2, 0.5, 1, 2, and 5 μg/mL for 48 h) compared with cisplatin-sensitive ESCC cells [59]. However, the half-maximal inhibitory concentration ($IC_{50}$) value of cisplatin treatment (Table 1) showed that *CCAT1* positively correlates with cisplatin resistance in ESCC cells. Importantly, *LINC01419* overexpression contributes to the diminished effect of 5-FU (10 μg/mL for 48 h) in ESCC cells by promoting the methylation of the promoter region of the glutathione S-transferase Pi 1 (*GSTP1*) gene [60] (Table 1). In addition to the lncRNAs mentioned above, *LINC00337* was overexpressed in ESCC patients' tissues and cell lines, which hindered the effects of cisplatin dose (0.5, 1, 2, and 3 μg/mL for 48 h) via the upregulation of TPX2 by recruiting the E2F4 transcription factor [51] (Table 1).

Furthermore, two lncRNAs, *MACC1-AS1* and *FOXD2-AS1*, were upregulated in ESCC cells and tumor tissues [61,62]. As a result, both the lncRNAs hinder cisplatin's efficacy (20, 40, 60, 80, and 100 µM) through the overexpression of NSD2 mRNA and protein in ESCC tissues compared to adjacent non-cancerous tissues [61] (Table 1). Moreover, *FOXD2-AS1* increases the cisplatin resistance (6.25, 12.5, 25, 50, and 100 µg/mL) by promoting the Akt/mTOR axis stimulation in ESCC cells [62] (Table 1).

Besides resistance to cisplatin, 5-FU, and paclitaxel, altered expression of lncRNAs resist the potential of gefitinib in ESCC treatment. For example, lncRNA *PART1* upregulated in gefitinib-resistant ESCC cells compared to parental ESCC cells. The resistance to gefitinib (0.01, 0.1, 1, 2, 3, 8, and 10 µM for 48 h) (Table 1) by lncRNA *PART1* is facilitated by the transportation of extracellular *PART1* into exosomes and incorporation into sensitive cells, which ultimately inhibits apoptotic proteins and cell apoptosis by regulating the Bcl-2/Bax pathway [63]. Moreover, upregulated *LINC01014* confers gefitinib resistance (1, 10, 20, and 30 µM for 48 h) (Table 1) in ESCC cells by inhibiting ESCC cells' apoptosis via PI3K/Akt/mTOR signaling pathway [18].

## 6. Conclusions and Future Aspects

In this review, we highlight the immense potential of oncogenic and tumor suppressive lncRNAs in regulating cancer-associated signaling pathways and their implication in drug resistance in ESCC patients. As mentioned in the previous sections, the Wnt/β-catenin and PI3K/Akt/mTOR pathway comprises multiple downstream signaling proteins, such as β-catenin, GSK3-β, Akt, PI3K, and mTORC1 complex, whose activation in association with dysregulated lncRNAs can manifest several hallmarks of cancer, including uncontrolled cell growth, inhibition of apoptosis, proliferation, increased metastasis, and invasion. We found that upregulated expression levels of *HERES*, *TUG1*, *HCP5*, and *PTCSC1* and downregulated expression levels of *UCA1* could be best suited for therapeutic application in clinical settings. These lncRNAs significantly target the key downstream molecules of cancer-related pathways, namely the Wnt/β-catenin and PI3K/Akt/mTOR pathway. Based on the ESCC cohort size and the detailed mechanism, *UCA1*, *HCP5*, and *PTCSC1* possess great potential as a therapeutic target for ESCC in association with the Wnt/β-catenin and PI3K/Akt/mTOR pathways, which signifies the clinical potential of lncRNAs for the treatment of ESCC patients. Thus, targeting lncRNAs and their associated pathways, i.e., Wnt/β-catenin and PI3K/Akt/mTOR, may provide novel approaches in the treatment and better management of ESCC patients.

We also presented the potential of lncRNAs as an effective regulator of chemotherapeutic drugs, such as paclitaxel, 5-FU, gefitinib, and cisplatin. In our opinion, upregulated levels of *AFAP1-AS1* could be a selective prognostic and therapeutic marker for the ESCC patients treated with cisplatin, 5-FU, and paclitaxel. Additionally, we observed that *AFAP1-AS1* lowers the efficacy of the maximum dose of cisplatin, 5-FU, and paclitaxel drugs at 24 h in 162 ESCC patients [58]. Therefore, the modulation of *AFAP-AS1* expression levels is of utmost importance in the management of ESCC. Furthermore, it has been shown that *LINC01014* resists the efficacy of gefitinib (30 µM at 48 h), which suggests that *LINC01014* requires critical attention concerning prognostic and therapeutic aspects in the clinical settings [18]. The diverse functional repertoire of lncRNAs reveals various opportunities for their therapeutic targeting, including inhibition at transcriptional and post-transcriptional levels, steric hindrance on protein interaction and formation of secondary structures, and the modulation of genomic loci or lncRNA expression patterns using clustered regularly interspaced short palindromic repeats (CRISPR)-associated proteins (Cas) technology. However, the application of RNA based therapeutics in clinical settings has been hampered by the lack of specificity, delivery method, and tolerability.

Besides, many clinical trials have shown the development of RNA therapeutics, such as miRNA mimics or antimiRs, and several are in phase II or III. Still, no lncRNA-based therapeutic agent has entered the clinical setting. In future, we believe that lncRNAs and their related signaling pathways can be targeted using secondary plant metabolites in com-

bination with chemotherapeutic drugs for the betterment of cancer treatment, as supported by recently published studies [64–66]. Experimental findings have indicated that bioactive phytochemicals, such as anacardic acid, baicalein, berberis, bharangin, genistein, calycosin, and silibinin, could be utilized to target the expression of lncRNAs in various cancers, including ESCC [64–66]. Additionally, it is likely that these bioactive phytochemicals may also modulate a diverse range of cell signaling pathways in cancer cells. Furthermore, novel formulations, such as nano-drug delivery systems, can be utilized to enhance the bioavailability of phytochemicals alone or in combination with chemotherapeutic drugs. In addition, synthetic chemistry tools may also be implemented to design new derivatives of existing drugs to analyze their potential to modulate lncRNAs in ESCC. Synergistic approaches may further enhance the activity of chemopreventive agents to optimize the levels of dysregulated lncRNAs.

Overall, our study provides comprehensive knowledge of the lncRNA regulated Wnt/β-catenin and PI3K/Akt signaling pathways and highlights the potential of lncRNAs hindering the therapeutic efficacy in ESCC. The techniques mentioned above can be employed to target desired lncRNAs clinically. However, future studies are required to translate the findings into clinical settings.

**Supplementary Materials:** The following are available online at https://www.mdpi.com/article/10.3390/curroncol29040189/s1, Table S1: Association of ESCC related lncRNAs in various signaling pathways. References [67–234] are referred to in Supplementary Materials.

**Author Contributions:** U.S. and A.J. conceived the original idea. U.S. wrote the manuscript and prepared the tables and figures. M.M. and H.S.T. wrote the conclusion section. T.S.B. formatted the references as per journal style. A.J., U.S., H.P., M.J., T.K. and A.B. contributed to the final editing of the manuscript. All authors have read and agreed to the published version of the manuscript.

**Funding:** Department of Science and Technology of India supported this work through the Indo-Russia grant (INT/RUS/RFBR/P-311) to AJ and DST-INSPIRE fellowship (IF180680) to U.S.

**Conflicts of Interest:** The authors declare no competing or conflicting interest.

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
