# Peer review of "A Pleiotropic Role of Long Non-Coding RNAs in the Modulation of Wnt/β-Catenin and PI3K/Akt/mTOR Signaling Pathways in Esophageal Squamous Cell Carcinoma: Implication in Chemotherapeutic Drug Response"

_curroncol, doi:10.3390/curroncol29040189_

Round 1

Reviewer 1 Report

In this article, the authors reviewed the role of lncRNAs in regulating the Wnt/β-Catenin and PI3K/Akt/mTOR Signaling in Esophageal Squamous Cell Carcinoma and associated chemo agents drug resistance. The authors put together a nice overview of the published studies along with good figures. This review article is helpful for the researchers and clinicians working with ESCC. Please address the following points:

-Please check for the consistency of space between the words. Many words have no space.
-Line 58- when the lncRNA is mentioned for the first time, give a little background about lncRNA and how it is different from other non-coding RNAs.
-Line 76- Indicate the new studies from 2022, if any, and in general, incorporate the references from 2022.
-Line 109- Correct the beta symbol (also elsewhere for GSK-3beta).
-Since lncRNAs, especially in ESCC, can have both tumor-suppressing and oncogenic roles, please discuss how could lncRNAs be targeted in the clinic, or lncRNAs are better suited for the prognosis of the tumor state and drug resistance only?

Author Response

The authors of this manuscript express their sincere thanks to the reviewer for the critical assessment of this work. The authors have acted upon the recommendations of the reviewer which have resulted in a significant enhancement in the quality of this manuscript. All modifications incorporated in the manuscript are highlighted in red color font. A “point-by-point” response to each and every comment is outlined below.

General comments:

In this article, the authors reviewed the role of lncRNAs in regulating the Wnt/β-Catenin and PI3K/Akt/mTOR Signaling in Esophageal Squamous Cell Carcinoma and associated chemo agents drug resistance. The authors put together a nice overview of the published studies along with good figures. This review article is helpful for the researchers and clinicians working with ESCC. Please address the following points:

Response:

Thank you for your expertise, time, and effort in reviewing our manuscript. We are deeply encouraged by the generous comments regarding the quality of our work. We sincerely appreciate the valuable suggestions which we have found extremely valuable while revising our manuscript.

Specific comments:

Comment 1:

Please check for the consistency of space between the words. Many words have no space.

Response:

Thank you for bringing this issue to us. We have now resolved this issue in the revised manuscript.

Comment 2:

Line 58- when the lncRNA is mentioned for the first time, give a little background about lncRNA and how it is different from other non-coding RNAs.

Response:

Thank you for the suggestion. We have mentioned the background of lncRNAs in the introduction section (page 2; lines 57-70).

Comment 3:

Line 76- Indicate the new studies from 2022, if any, and in general, incorporate the references from 2022.

Response:

Thank you very much for bringing up this issue.  We have included a relevant study from 2022 in the revised manuscript (page 2; line 64, reference number 24).

Comment 4:

Line 109- Correct the beta symbol (also elsewhere for GSK-3beta).

Response:

We have made the suggested corrections in the revised manuscript (page 4; line 115) and elsewhere.

Comment 5:

Since lncRNAs, especially in ESCC, can have both tumor-suppressing and oncogenic roles, please discuss how could lncRNAs be targeted in the clinic, or lncRNAs are better suited for the prognosis of the tumor state and drug resistance only?

 Response:

We admire the reviewer for this thought-provoking comment. We have now incorporated the suggested changes in the revised manuscript (page 14, line 31 to page 15, line 57).

Additionally,

  1. The reference list has been modified as we have added several new references. Special attention is given to conform to the order of references and bibliographic style of the journal.
  2. The supplementary table 1 has been revised with alterations marked with red color font.
  3. The entire manuscript has been thoroughly checked and edited to ensure uniform style, organization, and quality.

Finally, on behalf of my co-authors, I once again express my sincere thanks to the reviewer for the valuable suggestions and constructive input to improve the quality of our manuscript.

Reviewer 2 Report

Manuscript is very well written by the team, especially the figures. However, there are a few points that need to be addressed as below:

  1. Spacing issue “targetsvarious”, “moleculesof”, “LINC01014,block”, “includingcisplatin”, “gefitinib,used” in abstract should be corrected.
  2. Spacing issue in keywords “carcinoma;l ong non-coding”. Further, multiple instances of spacing issues were observed in text part of the manuscript also.
  3. In the line no. 43, it will be more appropriate if written as “the development of therapy counter-acting resistance mechanisms in tumor cells [3].”
  4. Chemotherapeutic resistance in cancer occurred mainly because of the mutations in the key genes. It will be appropriate, if authors will also include the information about the relevant mutations in lncRNAs.
  5. Conclusion and Future Aspects can be further detailed to give key home messages to the readers in a better way.
  6. References indicated in the manuscript are way much less than the references given in the supplementary Table. Authors could also discuss the references from supplementary tables in the main manuscript.

Author Response

The authors of this manuscript express their sincere thanks to the reviewer for the critical assessment of this work. The authors have acted upon the recommendations of the reviewer which have resulted in a significant enhancement in the quality of this manuscript. All modifications incorporated in the manuscript are highlighted in red color font. A “point-by-point” response to each and every comment is outlined below.

General comments:

Manuscript is very well written by the team, especially the figures. However, there are a few points that need to be addressed as below:

Response:

Thank you for your expertise, time, and effort for reviewing our manuscript. We are deeply encouraged by the generous comments regarding the quality of our work. We sincerely appreciate the valuable suggestions which we have found extremely valuable while revising our manuscript.

Specific comments:

Comment 1:

Spacing issue “targetsvarious”, “molecule”, “LINC01014,block”, “includingcisplatin”, “gefitinib,used” in abstract should be corrected.

 Response:

Thanks for your suggestion. We have now corrected in the revised manuscript as indicated.

Comment 2:

Spacing issue in keywords “carcinoma;l ong non-coding”. Further, multiple instances of spacing issues were observed in text part of the manuscript also.

 Response:

Thank you for your concern. We have taken care of this issue in the revised manuscript.

Comment 3:

In the line no. 43, it will be more appropriate if written as “the development of therapy counter-acting resistance mechanisms in tumor cells [3].”

 Response:

Thank you for your suggestion. We have now incorporated the suggested sentence in the revised manuscript (page 1; line 43).

Comment 4:

Chemotherapeutic resistance in cancer occurred mainly because of the mutations in the key genes. It will be appropriate, if authors will also include the information about the relevant mutations in lncRNAs.

 Response:

Thank you very much for your kind suggestion. Unfortunately, we could not find the literature having mutations/polymorphism on the listed lncRNAs in the context of ESCC.

 Comment 5:

Conclusion and Future Aspects can be further detailed to give key home messages to the readers in a better way.

 Response:

We have now discussed the Conclusion and Future Aspects in detail in the revised manuscript. (page 15, lines 58-63).

Comment 6:

References indicated in the manuscript are way much less than the references given in the supplementary Table. Authors could also discuss the references from supplementary tables in the main manuscript.

Response:

Thank you for the constructive view. We have now discussed the references cited in the supplementary Table 1 in the revised manuscript  (page 2; lines 77-85). We have also included all references from the supplementary Table 1 to the main manuscript.

 Additionally,

  1. The reference list has been modified as we have added several new references. Special attention is given to conform to the order of references and bibliographic style of the journal.
  2. The supplementary table 1 has been revised with alterations marked with red color font.
  3. The entire manuscript has been thoroughly checked and edited to ensure uniform style, organization, and quality.

Finally, on behalf of my co-authors, I once again express my sincere thanks to the reviewer for the valuable suggestions and constructive input to improve the quality of our manuscript.